# Discovering sparse control strategies in neural activity

**Edward D. Lee** [1] *, **Xiaowen Chen** [2], **Bryan C. Daniels** [3]

**1** Complexity Science Hub Vienna, Vienna, Austria, **2** Laboratoire de physique de l'École normale supérieure, CNRS, PSL Université, Sorbonne Université, Université de Paris, Paris, France, **3** School of Complex Adaptive Systems, Arizona State University, Tempe, Arizona, United States of America

* edlee@csh.ac.at

## Abstract

Biological circuits such as neural or gene regulation networks use internal states to map sensory input to an adaptive repertoire of behavior. Characterizing this mapping is a major challenge for systems biology. Though experiments that probe internal states are developing rapidly, organismal complexity presents a fundamental obstacle given the many possible ways internal states could map to behavior. Using *C. elegans* as an example, we propose a protocol for systematic perturbation of neural states that limits experimental complexity and could eventually help characterize collective aspects of the neural-behavioral map. We consider experimentally motivated small perturbations—ones that are most likely to preserve natural dynamics and are closer to internal control mechanisms—to neural states and their impact on collective neural activity. Then, we connect such perturbations to the local information geometry of collective statistics, which can be fully characterized using pairwise perturbations. Applying the protocol to a minimal model of *C. elegans* neural activity, we find that collective neural statistics are most sensitive to a few principal perturbative modes. Dominant eigenvalues decay initially as a power law, unveiling a hierarchy that arises from variation in individual neural activity and pairwise interactions. Highest-ranking modes tend to be dominated by a few, "pivotal" neurons that account for most of the system's sensitivity, suggesting a sparse mechanism of collective control.

## Author summary

The relationship between underlying biological circuitry and behavior is complex and difficult to probe experimentally. Part of the problem is that, for organisms of even modest size, there are an overwhelming number of possible combinations of interacting circuit components. We develop a theoretical framework to simplify this problem with experiments that change the system minimally with small perturbations. In the realm of small perturbations, not only does system behavior remain close to normal but the range of possible perturbations is greatly reduced to only pairs of experimental targets. We demonstrate such a perturbation using a minimal model of neural activity in the *C. elegans* worm. We find that a few combinations of "pivotal" neurons strongly affect the statistics of synchronous activity, suggesting they may be important for neural control of behavior.

**Data Availability Statement:** Source code for the analysis is available at https://doi.org/10.5281/zenodo.6412747.

**Funding:** We acknowledge funding from the Omega Miller Program (https://www.santafe.edu/research/initiatives/miller-omega-program),

National Science Foundation grant PHY1838420 (http://nsf.gov), and Bundesministerium Bildung, Wissenschaft und Forschung, HRSM 2016 (Complexity Science Hub Vienna) (https://www.bmbwf.gv.at). B.C.D. was supported by the ASU–SFI Center for Biosocial Complex Systems (https://complexity.asu.edu/asu-sfi). The funders had no role in study design, data collection and analysis, decision to publish, or preparation of the manuscript.

**Competing interests:** The authors have declared that no competing interests exist.

Our work suggests generalizable, feasible, and perturbative experiments to map how the physical components of an organism control emergent collective activity.

## Introduction

Control of complex dynamical networks is a problem of major interest in biology for finding drivers of diseased states [1–3] and determinants of behavior [4–7]. In neural systems, the question of which and how many neurons correspond to controllability is largely open. While there is evidence that single-neuron manipulation is sufficient to induce behavioral change in some cases [8, 9], other research shows behavioral information to be encoded amongst many neurons [4, 10, 11]. Formidable work has gone into highlighting specific behavioral circuits [12–14], but the general problem of identifying which neurons to probe experimentally and choosing how to perturb them is difficult: in principle, a thorough experiment would require a combinatorially large number of procedures to test all the possible ways that neural targets could be modified. Theoretical tools provide a way of winnowing down the number by using control-theoretic analysis of dynamical systems models [15, 16], structural analysis [5, 17–19], and network properties [20], amongst other approaches reviewed in reference [21]. Here, we develop a theoretical framework that explicitly considers perturbation experiments to discover candidate control neurons. Our framework suggests a way of leveraging recent developments in optogenetics [19, 22–27] that allow for fast, precise control of neurons and even the regulation of brain states with the closed-loop optogenetic [28–32] or optoelectronic systems [33].

Despite the generalization away from single neurons in the study of sensory encoding [10, 34], an echo of it still rings in studies of neural control of behavior [12]. Some experiments rely on the notion of individual (or sets of similar) "driver" neurons that—when ablated, silenced, or otherwise modified—substantially change behavioral outcomes [35–37]. While some individual cells, like interneurons connecting distal parts of the body, are essential for normal behavior, some higher-order behaviors are robust to the function of individual cells [14, 38–40]. Indeed, the finding that some neural circuits encode sensory input in a sparse, distributed way raises the question of whether or not neural control follows similar principles [4, 41–43].

We explore two potential sources of sparsity in neural control of collective activity. First, a desired output could be equally well produced by many possible changes, or modes, at the neural level or by only one particular change. If we think of neural states crudely as parameter settings and a mode consists of changing all the parameters in a preset way, this is asking if many such presets matter. We denominate cases where there is a clear separation in the importance of modes as "sloppy" and where there is no such separation as "non-sloppy" [44]. Second, each mode can itself be sparse or dense in the number of neurons involved [45–47], corresponding to the complexity of coordinating multiple neurons simultaneously. These possibilities lead to the four hypotheses we draw in Fig 1, where control is sloppy or non-sloppy with either sparse or dense collective modes of sensitivity (see Appendix A in S1 Text for theoretical overview).

In the model organism *Caenorhabditis elegans*, the hypothesis that neural control is sparse is supported by observations that both neural and behavioral dynamics live on sparse manifolds. Neural activity displays collective modes originating from circuit components or global encoding of information that effectively reduces its dimensionality [5, 17, 48, 49]. Complementing this, worm posture is low-dimensional, and four eigenmodes capture nearly all the variance in shape space [42, 50]. More recent work combines the two types of measurement to show that dynamical embedding of worm locomotion along dimensions of neural activity extracts a few principal neural dimensions that capture transitions between behaviors [51, 52]. In some

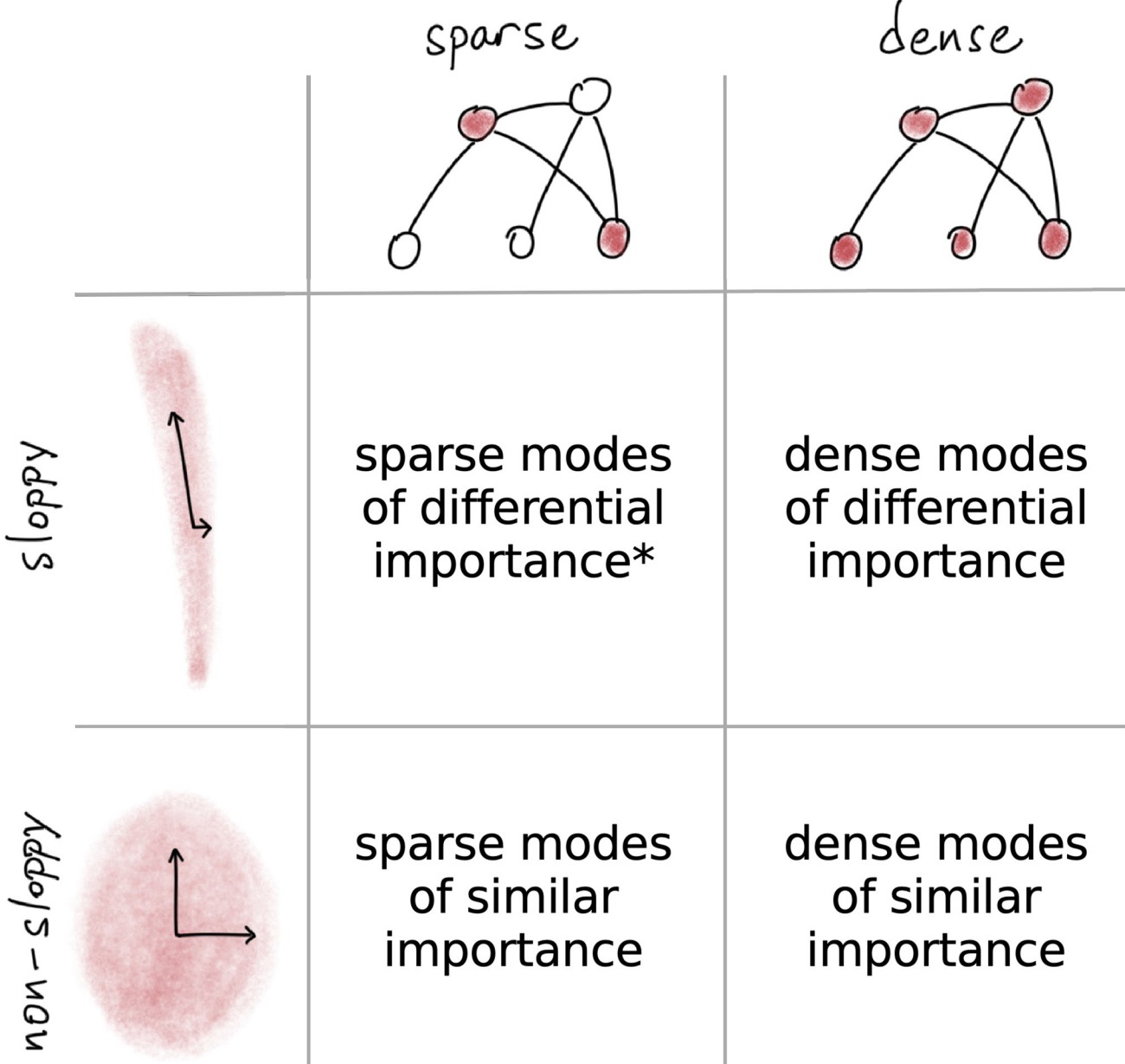

**Fig 1. Four possibilities for sparse sensitivity.** When collective activity shows sloppy structure, the local information geometry is elongated along sloppy, insensitive directions. With dense collective encoding, multiple components each matter equally. Starred combination with sloppy, sparse combinations of neurons aligns with centralized control.

cases, neural modes are even predictive of longer-term behavioral regularities [53]. Though neural signals may not simply reflect behavioral control, these findings support the hypothesis that common mechanisms of neural control may underlie neural-behavioral similarities between individuals. Inspired by signatures of sparseness in neural-behavioral mapping, we examine sparseness in collective neural activity response to individual neurons as a step towards considering behavior.

To test these hypotheses is difficult because an exhaustive protocol for probing subsets of neurons, even for a minuscule and precisely mapped organism such as *C. elegans*, is impossible

[54]. A combinatorially large number of possible tests could involve simultaneous perturbations to subsets of neurons in ways that are sympathetic, antagonistic, and of varying magnitude. Could we systematically analyze all such combinations of perturbations in a feasible way?

Our first result is to show one way in which such complexity can be dramatically reduced. By focusing on the limit of small perturbations, and given the remarkable effectiveness of pairwise models of neural activity [46, 48, 55], we demonstrate how experiments that perturb only pairwise correlations could be enough to fully characterize the local information geometry.

Our second result comes from demonstrating how this simplification could be useful in the context of an *in silico* experiment, making use of data on *C. elegans*, a particularly interesting example with which to explore neural control. Current experiments measure whole-brain activity in the freely moving worm and soon will allow matching neurons to the structural connectome [56–58]. Combined with tools for manipulating neural states [32], our proposed experimental protocol could be used to search for neural centers important to collective activity.

In the following text, we start by developing the basis for our approach in "Methods and theoretical formulation." We cover minimal statistical models for neural activity and relate theoretical model perturbations to realistic experimental ones, focusing on a measure of neural collective behavior. Afterwards, we apply the described approach to data sets that display signatures of sparse control in "Results in *C. elegans*." At the end, we discuss our findings and questions of experimental feasibility.

## Methods and theoretical formulation

### Maximum entropy (maxent) model

Our theoretical framework begins with a minimal statistical model of neural activity. In the case of spiking neurons, this could be a pairwise maximum entropy model specifying statistics of activity and inactivity. In the case of *C. elegans* that we focus on here, we distinguish between its more gradual change in membrane voltage in contrast with the rapidly spiking neuron of the mammalian brain. Though we might then think to use $Ca^{2+}$ levels as a measure of neural state, we find that fluorescence measurements of absolute levels of calcium are dominated by noise, but it is possible to extract the derivative reliably such that it reflects real correlations in neural activity [48]. Given instrumental limitations that prevent us from measuring the precise continuum dynamics, we focus on a minimal discretization of the time series that still captures the dominant tendencies in the data, coarse-graining each neuron $m$'s derivative as up $s_m = 1$, down $s_m = -1$, flat $s_m = 0$. Whereas a binary discretization would make it impossible to code inactive neurons, a finer discretization leads to exponentially increasing computational costs with the number of states. The optimal procedure should be sensitive to the experimental protocol, but the practical principle of minimal and sufficient description of neural activity leads us to a ternary description of *C. elegans* neural states.

With this discretization procedure, we denote the time-averaged probabilities of being in state $k$,

$$r_k(s_m) = \frac{1}{T}\sum_{\tau=1}^{T} \delta_{s_m,k}(\tau),$$

(1)

where the Kronecker delta function $\delta_{s_m,k}$ indicates when the state of the neuron $s_m$ is $k$ at time $\tau$ over the duration of the experiment $T$. Similarly, the pairwise probabilities of agreement between any two neurons

$$\langle s_m s_t \rangle = \frac{1}{T}\sum_{\tau=1}^{T} \delta_{s_m,s_t}(\tau),$$

(2)

and higher-order correlations describe the probabilities of agreement between multiple neurons. Higher-order correlations are not necessarily given by the lower-order statistics, but only accounting for pairwise correlations is sufficient to capture higher-order correlations [48] (Appendix B in S1 Text).

The maxent approach captures statistics of neural activity in a way that can reflect latent physical interactions while also obeying a quantitative formulation of Occam's Razor [60, 61]. This is encoded in the maxent principle that a model of the probability distribution $p(s)$ only match specified constraints but otherwise remain as structureless as possible. This can be done by maximizing the model's information entropy, $S = -\sum_s p(s) \log p(s)$, here a sum over all $3^N$ possible configurations, with the standard method of Langrangian multipliers [46, 62–64]. When the set of average individual neural activity $\{r_k(s_m)\}$ is constrained, the maxent procedure gives the "independent model," but it fails to capture correlations in neural activity as we show in Fig 2 (also see Fig B in S1 Text).

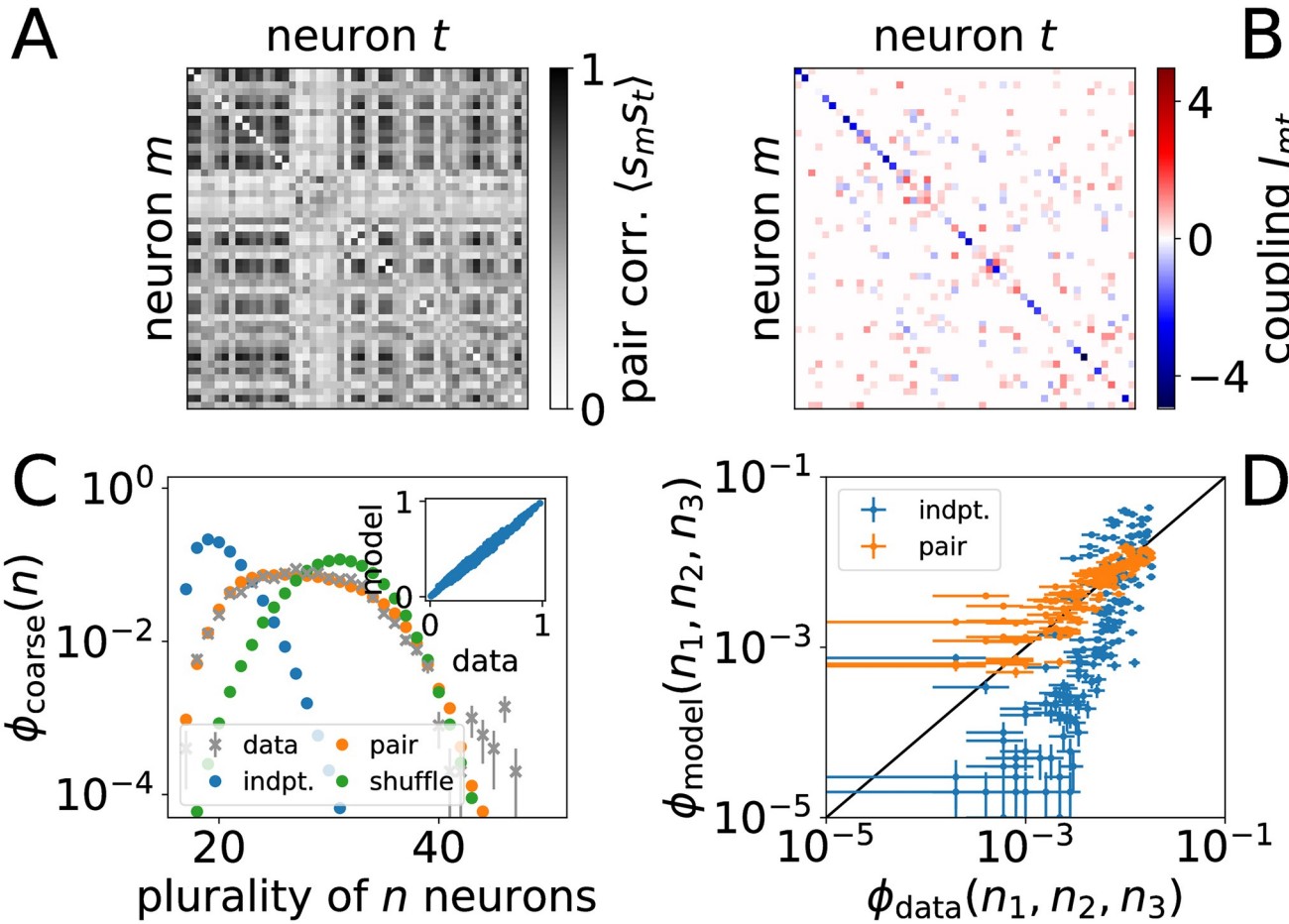

**Fig 2. Pairwise maxent model of anterior neural activity from reference [59] (see Fig B in S1 Text for another experiment).** (A) Pairwise correlations between subset of $N = 50$ neurons. Average individual neuron $r_{k=-1}(s_m)$ shown along diagonal. (B) Inferred biases $h_{m,k=-1}$ along diagonal and matrix of couplings $J_{mt}$ off the diagonal. (C) Coarse collective synchrony, probability that a plurality of $n$ neurons coincide $\phi_{\text{coarse}}(n)$, for data, pairwise maxent model, independent model, and shuffled couplings. Inset compares pairwise correlations for data with maxent model. Error bars show one standard deviation over bootstrapped samples. (D) Fine-grained synchrony $\phi_{\text{fine}}$ in the independent vs. pairwise maxent model. This is the probability of set sizes, the number of neurons in each of the three states when ordered $n_1 \geq n_2 \geq n_3$ such that $n_1$ corresponds to the size of the plurality and $n_3$ the smaller minority. Error bars show one standard error.

A simple variation that does not assume independence involves constraining the pairwise correlations from Eq 2. Given the finite-size of the data set, we constrain the pairwise correlations to obtain the pairwise maxent model with distribution

$$
\begin{aligned}
p(s) &= \exp[-E(s)]/Z \\
Z &= \sum_s \exp[-E(s)] \\
E(s) &= -\sum_{m<t}^{N} J_{mt}\delta_{s_m,s_t} - \sum_{k=-1}^{1}\sum_{m=1}^{N} h_{m,k}\delta_{k,s_m}.
\end{aligned}
\tag{3}
$$

The function $E(s)$ is known as the "energy." The log-probability of configuration $s$ varies with its energy such that lower energy means larger probability. As with the independent model, increasing neural bias $h_{m,k}$ pushes neuron $m$ to state $k$ while increasing the magnitude of coupling $J_{mt}$ magnifies the tendency of neurons $m$ and $t$ to agree when the coupling is positive (or to disagree when negative). Though the couplings are in principle exactly specified by the pairwise correlations in the data, sampling and experimental noise mean that there are many possible sets of couplings that align within the statistical variation of the data sample.

We focus on a numerical solution that recovers topological structure in the coupling network and has been shown to faithfully capture collective neural patterns [48]. We show the coupling network in Fig 2B. The two examples we consider are of $N = 50$ neuron subsets from the neurons located in the anterior of the immobilized worm from reference [59]. The solution then defines a minimal interaction network, distinct from the pattern of pairwise correlations, that characterizes dependencies in neural activity across the timescales represented in the data available.

## Mapping perturbations from experiment to simulation

In a hypothetical experiment, we might effect perturbations by inserting electrodes into immobilized worms or, in a less invasive and more natural protocol, by using optogenetic tools—by clamping membrane potential and effectively coupling neurons to one another via an external circuit to control the strength and timing of perturbation as is pictured in Fig 3A. Here, we use a perturbation that with small probability copies the state of one neuron to another, chosen to resemble increase or decrease in synaptic strength. Specifically, we select pairs, identifying a "target" neuron in state $s_t$ and a "matcher" neuron in state $s_m$, and clamp the matcher to the target's state, $s_m = s_t$, with small probability $\epsilon \ll 1$ as in Fig 3B. Compared to more common clamping and ablation techniques [31, 65], this proposed closed-loop perturbation is not as drastic, mostly preserves natural dynamics and is closer to internal control mechanisms such as synaptic modulation that do not require constant input from the outside [66–68].

Then, the modified probability $\tilde{r}_k$ that the matcher neuron is in state $k$ is the mixture

$$
\tilde{r}_k(s_m) \equiv (1 - \epsilon)r_k(s_m) + \epsilon\, r_k(s_t).
\tag{4}
$$

As a result of the perturbation in Eq 4, the statistics describing neuron $m$ becomes more like those of neuron $t$. This procedure likewise modifies the matcher neuron's coincidence $p(s_m = s_j) \to \tilde{p}(s_m = s_j)$ with all other neurons indexed $j$ as if modifying synaptic weights [47]. If we were to take sufficiently large $\epsilon$ in experiment, we would expect such intervention not to remain localized to neuron $s_m$ but to alter the neighbors and eventually the neighbors of neighbors and so on. If $\epsilon$ is small, then we expect a localized perturbation to be able to approximate well the desired change in statistics given in Eq 4. By taking the limit where the perturbation is

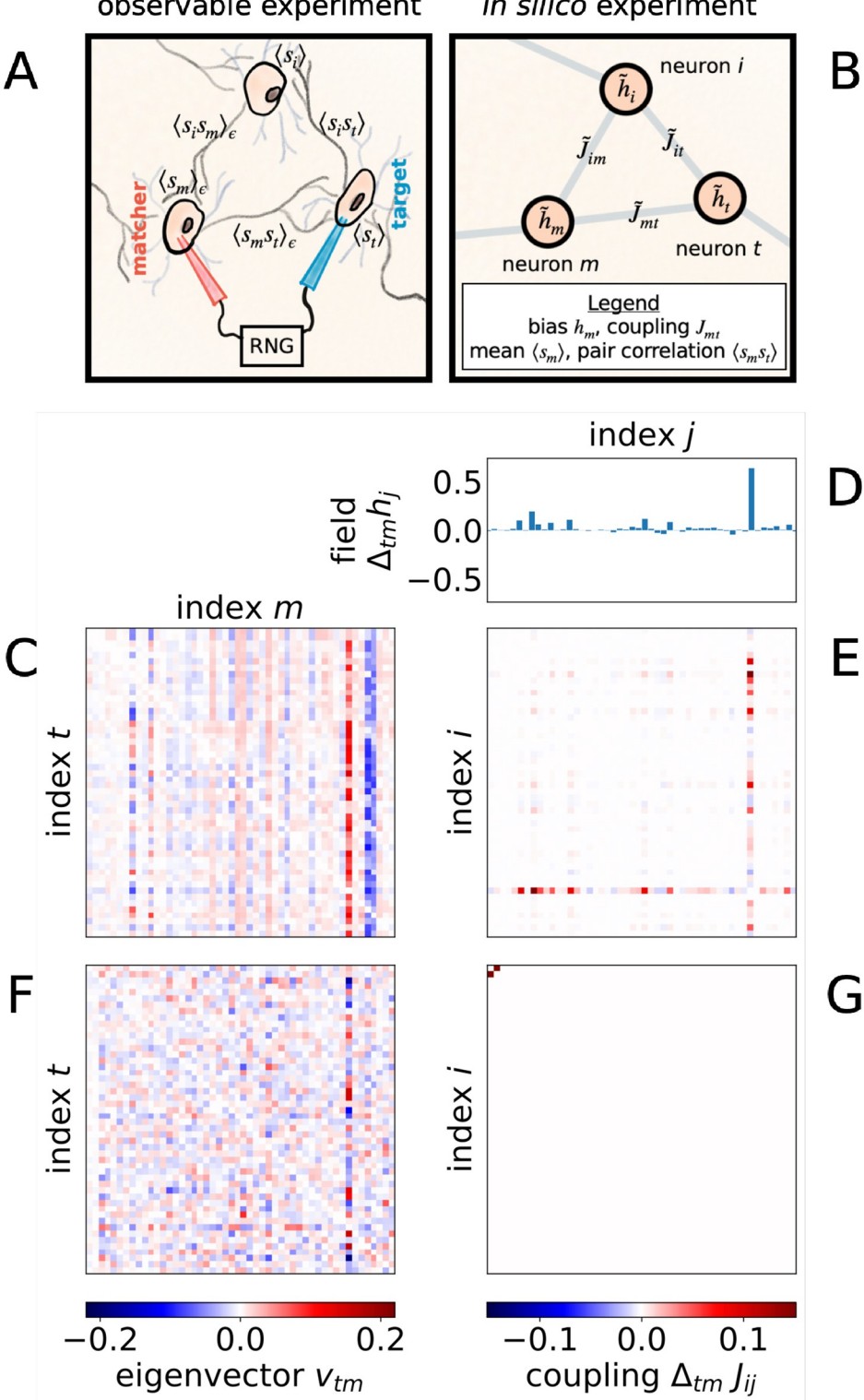

**Fig 3.** (A) Perturbation thought experiment consists of clamping matcher neuron $m$ to the state of target neuron $t$ with some small probability $\epsilon$ when indicated by a random number generator (RNG). We draw electrodes controlling membrane voltage, but optogenetic protocols are more elegant. (B) Perturbation corresponds to modifying fields and couplings in a pairwise maxent model. (C) Principal eigenmatrix for fine-grained synchrony mapped to change in (D) biases $\{h_i\}$, shown only for the "down" state, and (E) couplings $\{J_{ij}\}$ for $\epsilon = 10^{-4}$. (F) Diffuse observable perturbation using replacement rule from Eq 4 corresponds to (G) localized "natural" perturbation to one coupling (note nonzero values in the top left corner).

sufficiently small and assuming that it then remain localized, we specify a perturbation that corresponds to a unique change of parameters.

Such uniqueness implicitly depends on constraining ourselves to considering perturbations that move us along the family of pairwise maxent models. This constraint is not reflected in the replacement rule in Eq 4, which is compatible with an infinite variety of changes to higher-order correlations. In other words, we do not necessarily need to restrict ourselves to considering only the energy function defined in Eq 3, but we could allow for higher-order interactions to appear under perturbation. This introduces ambiguity that can only be resolved by choosing the form of the perturbations. Once we have chosen to fix the structure of the energy function from the maxent principle, however, we do not allow a perturbation to arbitrarily alter the form. This assumption is consistent with the widespread observation that the pairwise model generally captures well collective features of biological neural networks of modest size [46, 69–73]. Thus, we use the pairwise maxent model not only to specify a minimal, compressed representation of neural statistics, but also to specify how the probability distribution evolves under the replacement rule.

The replacement rule maps perturbations on observed statistics to system sensitivity instead of those on model parameters as is often the first instinct for theorists (see Appendix F in S1 Text). In the latter formulation, the parameters of the maxent model such as fields and couplings form the basis for "natural" or "canonical" perturbations [74]. This perspective adheres to the physical intuition that the parameters in the energy function reflect latent physical interactions. When the pairwise maxent model instead serves as a phenomenological approximation of the underlying physical interactions, a natural perturbation may be mediated by unknown parameters instead of fields and couplings [55, 75]. This obscures the intuition that changing fields should alter the individual neuron biases while couplings should alter the tendencies of pairs to agree.

Partly for this reason, the straightforward model perturbation to couplings in Fig 3B becomes much more complex when translated into its reciprocal experiment in Fig 3A. We show an example of a set of neural perturbations in panel C, indicating roughly uniform perturbations to three dominant neurons, that is more complex in the reciprocal space of fields and couplings in panels D and E, respectively. Again reflecting this basic principle, a trivial coupling perturbation in panel G is complex in the space of matcher-target perturbations in panel F. Additionally, observable perturbations do not depend on the model used but are preserved regardless of how a model is parameterized, ensuring consistency when our framework is extended to other model classes. Thus, defining perturbations in observable terms as in Eq 4 allows for easier experimental interpretation and for consistent comparison of inferred models [47].

## Neural synchrony for collective activity

If it is collective neural activity that encodes behaviorally relevant information [45, 46], then we can use collective properties as a proxy for behavior. As one measure of collective activity, we consider the probability that there are $n_1 \geq n_2 \geq n_3$ neurons in each of the three states, $\phi_{\text{fine}}(n_1, n_2, n_3)$, which we call a fine-grained measure of neural synchrony. We define $n_1$ to correspond to the size of the plurality and $n_3$ the smaller minority (i.e. there is no fixed correspondence to "rise," "fall," and "flat"). While synchrony does not differentiate between the orientation of the states, it presents a simple and computationally tractable statistic [76, 77], can contain information about worm posture and velocity (see Appendix C in S1 Text), and is reproduced by the pairwise maxent model as we show in Fig 2D. For comparison, we also consider $\phi_{\text{coarse}}(n_1)$ that only considers the number of neurons in the plurality in Fig 2C. We show

additional results using coarse synchrony in the Appendices. Importantly, synchrony treats neurons on equal footing with one another, and the symmetry ensures that neurons are distinguished by bias and variation in their local interaction networks and not from our measure of collective activity.

Under perturbation of a pair of matcher and target neurons, the synchrony distribution $\phi$ is mapped to an altered $\tilde{\phi}$ that depends on the strength of perturbation $\epsilon$. To measure how quickly the distribution changes with the perturbation, we use a unique characterization of the distance between distributions, the Kullback-Leibler divergence $D_{\mathrm{KL}}$. In the limit of infinitesimal perturbation, $\epsilon \to 0$, this becomes the Fisher information. Recognizing that only the first nonzero term is the second derivative as detailed further in Appendix A in S1 Text, we end up with the matrix of second derivatives, the Fisher information matrix (FIM) with respect to a matcher-target neuron pairs $mt$ and $m't'$,

$$
\begin{aligned}
F_{mt,m't'} &\equiv \lim_{\epsilon \to 0} \frac{2}{\epsilon^2} D_{\mathrm{KL}}\big[\phi||\tilde{\phi}\big] = \lim_{\epsilon \to 0} \frac{2}{\epsilon^2} \sum_{n_1 \geq n_2 \geq n_3} \phi \log\left(\frac{\phi}{\tilde{\phi}}\right) \equiv \lim_{\epsilon \to 0} \frac{2}{\epsilon^2} \left\langle \log\left(\frac{\phi}{\tilde{\phi}}\right) \right\rangle \\
&= \sum_{\theta_i,\theta_j} \left\langle \frac{\partial^2 \log\phi}{\partial\theta_i\theta_j} \, \mathcal{J}^{ij}_{mt,m't'} \right\rangle,
\end{aligned}
\tag{5}
$$

where the Jacobian $\mathcal{J}^{ij}_{mt,m't'}$ transforms the maxent parameters such as fields and couplings $\{\theta_i\}$, here ordered along a single index $i$, into the vector of changes that correspond to the observable perturbations specified in Eq 4. Eq 5 measures the impact of perturbation on a coarse-grained probability distribution, the collective synchrony, and accounts for the multiplicity of microscopic configurations belonging to a single coarse-grained state (Appendix A in S1 Text).

The FIM, whose entries are defined in Eq 5, is spanned by eigenvectors that describe orthogonal perturbations of the parameters $\{\theta_i\}$, where modes with large eigenvalues describe perturbations to which collective synchrony is highly sensitive and small eigenvalues represent ones to which the system is insensitive [47, 78]. Importantly, the basis vectors can involve a mix of antagonistic and sympathetic perturbations of neurons of varying magnitude, one that would be nontrivial to extract *a priori*. Thus, the FIM encodes how quickly coarse-grained configurations $\phi$ change as we modify the system on a microscopic level by perturbing pairs of neurons at a time, with perturbations that mimic changes to synaptic connectivity.

## Results in *C. elegans*

### Leading FIM eigenvalues show Zipfian decay

In Fig 4A, we show the rank-ordered eigenvalue spectrum of the FIM for collective synchrony $\phi_{\mathrm{fine}}$ calculated with the pairwise model in comparison with several null models: independent neurons, pairwise maxent model with couplings randomly shuffled between all pairs of neurons (which preserves the distribution of couplings but not the topology of the interaction network), and canonical perturbations directly modifying each coupling at a time by a fixed amount. Across all cases, we expect a hard cutoff at rank $Z_{\max} = 234$, the dimensionality of the synchrony distribution $\phi_{\mathrm{fine}}$. In contrast with the others, the independent neuron model has short maximum cutoff well below the theoretical maximum, reflecting the essential role of interactions in mediating pairwise perturbations. The cutoff for the pairwise maxent model reveals the replacement rule in Eq 4 to be sufficient to nearly span the full dimensionality of synchrony space. This observation confirms that the pairwise model with the pairwise replacement rule generates an appropriate basis with which to explore collective sensitivity.

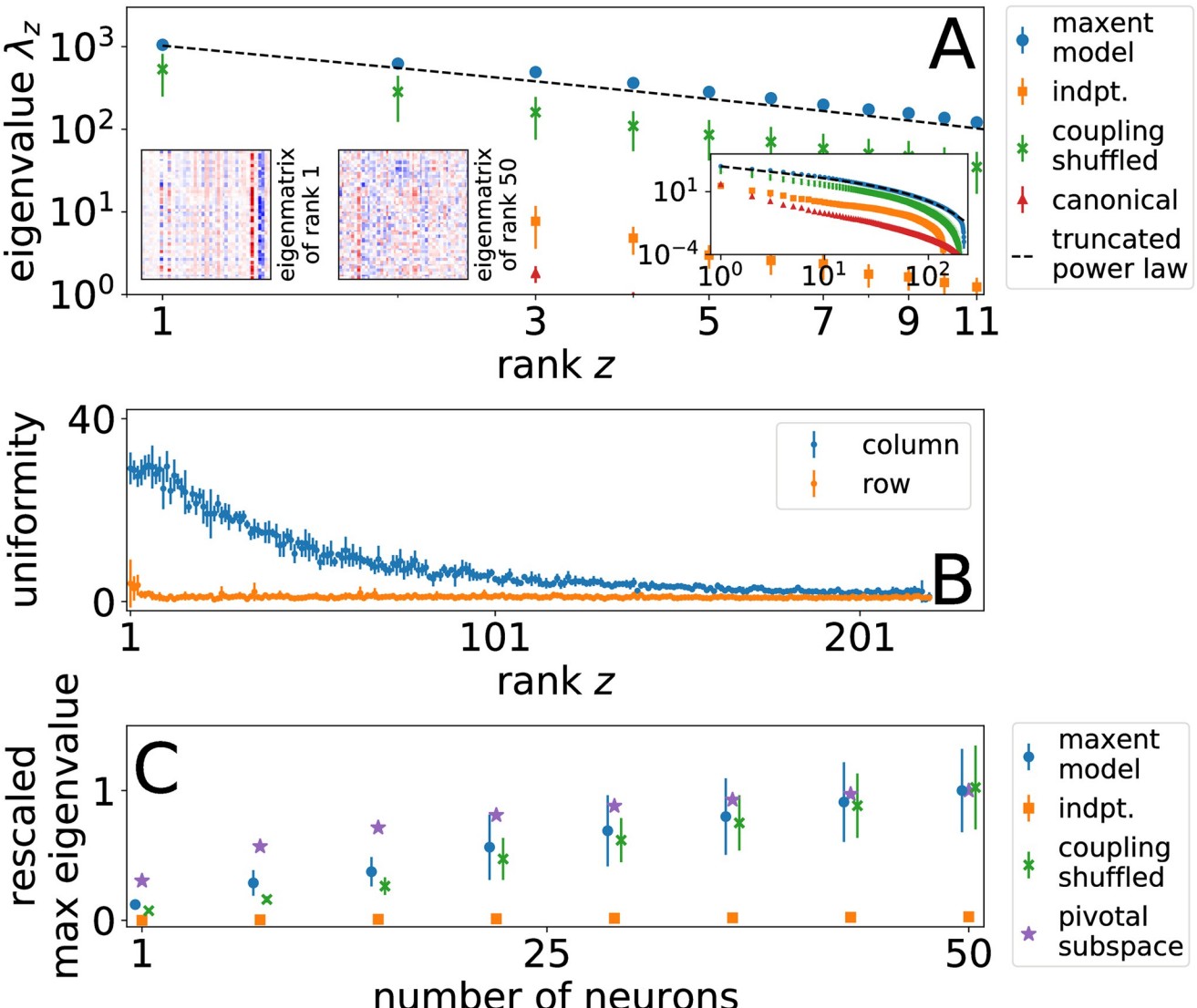

**Fig 4.** (A) FIM eigenvalue spectrum for pairwise maxent, independent (indpt.) and shuffled couplings null models. Results are averaged over $M$ Monte Carlo samples ($M = 10$ for maxent model and random shuffles, $M = 4$ for indpt. and canonical). For comparison, we show response to canonical perturbation to couplings. Insets on left show example eigenmatrices of rank 1 and 50, but only the first displays strong vertical striations. Inset on right shows full eigenvalue spectrum. Error bars show standard error of the mean. (B) Eigenmatrix column and row uniformity. Error bars represent a standard deviation across Monte Carlo samples. (C) Rescaled sensitivity. Principal FIM eigenvalues but for neuron subsamples as a function of subsample size. Normalized by the average maximum eigenvalue for the maxent model. Error bars represent standard deviation around means of Monte Carlo samples. Points have been offset for visibility along the x-axis. Compare with Fig P in S1 Text.

The rank-ordered spectrum of eigenvalues $\lambda_z$ of rank $z$ initially decays such that each successive level of perturbation returns multiplicatively smaller response, following on limited range Zipf's law, $\lambda_z \propto z^{-1}$. In contrast, a simple exponential decay would indicate a sharp cutoff for sensitive modes beyond some rank. We find that exponential decay alone does not describe the eigenvalue spectrum and instead find a reasonable fit using a power law with an exponential tail,

$$\lambda_z = A z^{-\alpha} e^{-z/\bar{z}}; \quad z \leq Z_{\max}, \tag{6}$$

which includes the exponential form as the special case when $\alpha = 0$. In Eq 6, we have parameters for vertical scaling $A$, exponent $\alpha$, tail location $\bar{z}$, and numerical precision cutoff $Z_{\max}$ (see

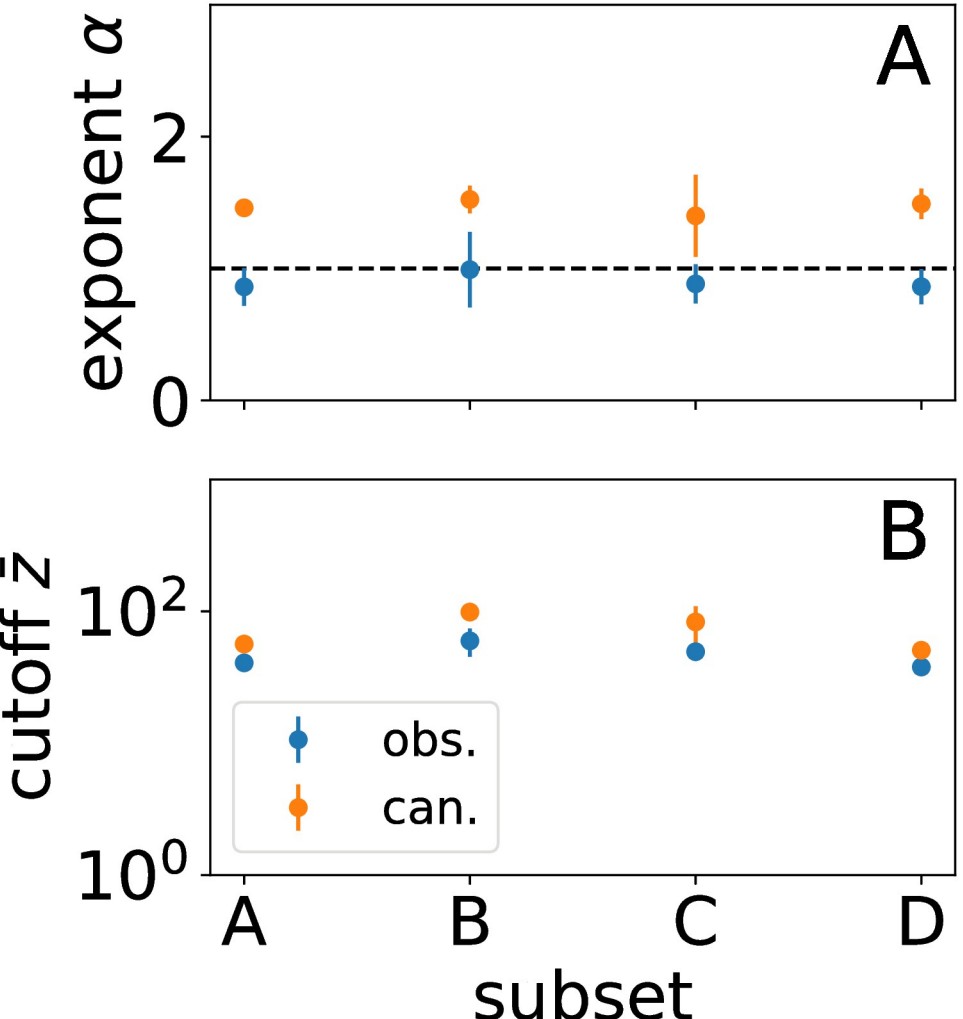

**Fig 5.** (A) Power law exponent from fitting FIM eigenvalue spectra comparing observable and canonical perturbations. (B) Exponential tail cutoff $\bar{z}$. See Fig F in S1 Text for another experiment.

Section E of S1 Text for more details about the fitting procedure). We find that the truncated power law usually presents a compelling fit, and the scale of the cutoff $\bar{z} \gtrsim 30$ indicates a substantial power-law regime as graphed in Fig 5.

Scaling seems to be a general feature of system statistics because it occurs both in the maxent model and the nulls. Notably, the exponent for canonical decay is faster than that from observable perturbations. The shuffled couplings model, however, is much closer than the others to the scaling for the pairwise maxent model of the data, suggesting that the topology of the interaction structure is not primarily determining sensitivity but that sensitivity depends on statistical properties of the coupling distribution. This is evocative of theoretical results analyzing the stability of neural networks [79]. Thus, the statistical hierarchy reflects the role of component disorder common across the models whose overall collective sensitivity is magnified by interactions.

On the other hand, the vertical scales and the scaling exponents—which we find to be close to $\alpha = 1$ for the maxent model as in Fig 5—vary amongst the nulls. Unsurprisingly, the spectra are overall scaled lower for the independent model since any particular perturbation is isolated to that neuron's contribution to synchrony. The spectra for canonical perturbations also are

generally lower, but we expect perturbations in the space of observable and model parameters to be scaled differently from the transformation of variables (Appendix G in S1 Text). In short, the null models we consider (aside from the shuffled couplings) are distinguishable from the observable perturbation in the eigenvalue spectrum.

## FIM eigenvectors reveal pivotal neurons

Inspecting the corresponding eigenvectors, we find some modes are dominated by perturbations focused only on a few neurons. To better represent the connection between pairwise perturbations and eigenvectors, we reshape eigenvectors into *eigenmatrices* $V_{mt}$ such that the elements in a column indicate how matcher neuron $m$ should imitate its target neighbors $t$ in turn. When put into this representation, the eigenmatrices display vertical striations as in the inset of Fig 4A. These striations are visible because they are almost exclusively of the same sign, indicating that the mode describes perturbations localized to a single neuron that tend to increase or decrease its correlation with all neighbors simultaneously. In contrast, horizontal striations that represent uniform perturbations across all the neighbors of a particular neuron, a kind of global perturbation, tend to be sparser and weaker on average. This emergent pattern contained in the block structure of the FIM suggests that localized, uniform enhancement or suppression of synaptic connections leading to a small set of *pivotal* neurons may serve as effective mechanisms for modulating collective activity.

As a more direct analysis of pivotal neurons, we limit our analysis to perturbations focused on a single matcher neuron at a time. When the FIM is ordered first by matchers and then targets for each matcher, these perturbations correspond to blocks along the diagonal of dimension $(N-1) \times (N-1)$ for fixed $m$ and variable $t$ [47]. From the principal eigenvalues of the diagonal blocks, we find that pivotal neurons with the largest eigenvalues tend to coincide with the ones that manifest in vertical striations of the full FIM (Appendix E in S1 Text). These striations correspond to columns with high uniformity.

As a measure of this, we define row and column uniformity, respectively,

$$
\begin{aligned}
U_i &\equiv \left( \sum_{j=1}^{N-1} v_{ij} \right)^2 \\
V_j &\equiv \left( \sum_{i=1}^{N-1} v_{ij} \right)^2,
\end{aligned}
\tag{7}
$$

for each eigenmatrix $v_{ij}$ with unit norm. When we consider the subspace of leading pivotal neurons and compare them with the subspace of randomly chosen neurons, the principal eigenvalues of the former set saturate much more quickly as shown in Fig 4C, indicating that there is a subset that dominates collective sensitivity.

The patterns that we note both in the eigenvalue spectrum and eigenvector basis hold generally for random neuron subsets of $N = 50$ sampled from each experiment, the maximum possible number that can be modeled without overfitting [48]. In contrast with the typical notion that the identities of particular neurons are essential, many pivotal neurons fluctuate between subsets and random samples as shown in Fig 6. This means that the appearance of pivotal neurons is a feature of the ensemble statistics, a point that we confirm with the shuffled couplings null (Fig I in S1 Text). Since collective synchrony is a lower-dimensional statistic and in principle permits exchange symmetries between neurons, this is not necessarily surprising. On the other hand, we note that these collective properties are robust to heterogeneity in network connectivity, bias, and interactions that would tend to break such symmetries.

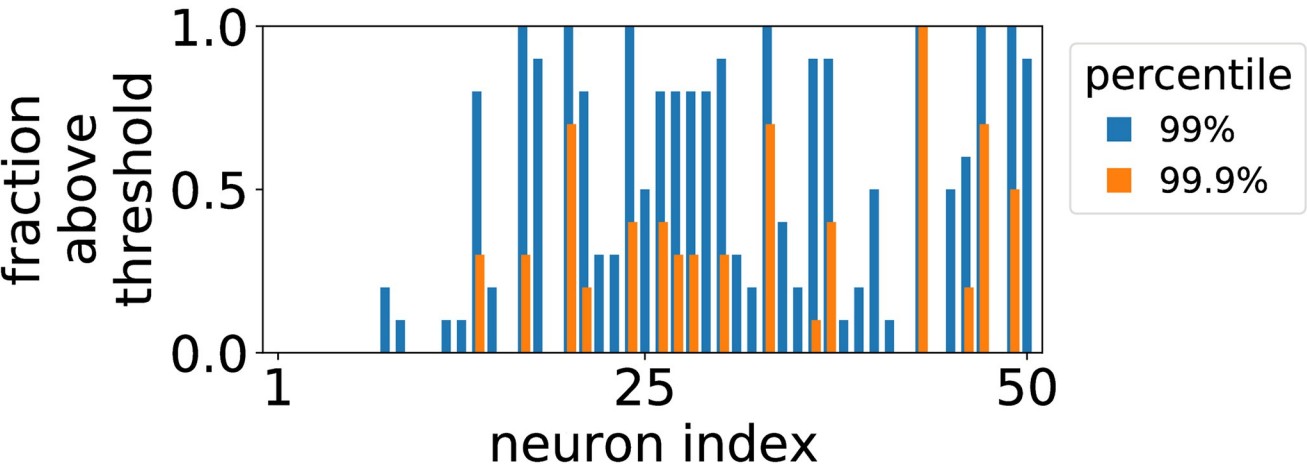

**Fig 6. Fraction of times out of 10 Monte Carlo samples that neuron column uniformity exceeds 99% and 99.9% percentile cutoffs out of column uniformities over all neurons and eigenmatrices.** See also Fig S in S1 Text.

## Discussion

Examples of how sensory and behavioral information is encoded in neural activity suggest that while information is sometimes localized to a few neurons it is at other times distributed amongst many. This may result from information flow between collective to individual components and because the precise scale at which information is processed may vary with time, function, and organism [80]. We develop a perturbative approach that is sensitive to the potential variety of involved neural scales. Using an information-geometric perspective, we identify statistical aspects of the distribution of neural activity that are essential for preserving collective activity. Importantly, we do not have to assume that collective properties will be sensitive to individual neurons or certain combinations but can discover the appropriate ones in a principled way. It becomes feasible to map out the combinations comprehensively because the response of the system depends on only pairs of small perturbations, a property of analyticity of the mapping from activity to collective state.

As an *in silico* realization of the protocol, we use perturbations that mimic internal neural coordination and calculate their impact on collective synchrony. In a minimal model, we find that dominant perturbative modes do not tend to be distributed amongst many neurons but are localized into pivotal ones. The concentration of sensitivity in a few neurons is analogous to the presence of driver neurons, modulation of which drives the system from one configuration to another [81–83]. Interestingly, we find that pivotal neuron identities fluctuate across subsets and Monte Carlo samples used to estimate the entries of the FIM. Instead of specific pivotal neurons, it is the collective properties of localized eigenvectors and scaling in sensitivity that are preserved.

The collective properties require neural heterogeneity. Since identical neurons would imply uniform eigenvectors, variation in each neuron's local interaction network and bias is responsible for the emergence of pivotal neurons. By shuffling the inferred interaction network, we verify that these features seem to be a result of the distribution and magnitude of interactions but not the specific topology. In this sense, the neurons do not need to be labeled differently, but the collection of local interactions and biases is responsible for the statistical properties of the FIM, echoing findings about statistical properties that make neural networks stable [79].

At first glance, this seems to be at odds with the consensus that neurons involved in motor circuits play fixed roles such as AVAs in reversals [4]. It is not. We focus on collective

synchrony as a simple proxy for behavior, and interchangeability of pivotal neurons may stem from the way that synchrony treats each neuron equally. By definition, it cannot distinguish between neurons with respect to a particular behavior. If we instead imagine searching for controllers of specific behaviors, this symmetry is likely lost such that control may be localized in fixed neurons. It remains to future work to reconcile our result about a statistically robust property of collective neural activity and its functional implications for sparse behavioral control.

One major question that we approach in our framework is how experimental perturbations should be represented in the model. While the focus is often on model parameters as representations of physical dials that are experimentally accessible, direct correspondence is not obvious for a statistical model inferred from data. As an example, when couplings inferred by a pairwise maxent model were directly compared with physical contact between protein residues, the model recovered only a subset of real interactions even while recovering the statistical ensemble [60, 84]. Taking the pairwise maxent model, it is clear exactly what increasing a coupling does to the energy function, enhancing the tendency of a pair of neurons to coincide, but the actual result is nontrivial modification of the entire distribution. For the experimentalist, it seems more natural to consider the problem from the perspective of the observable statistics that can be perturbed in a controlled and precise manner. In the case of Boltzmann-type models, the relationship between observables and model parameters can be made exact for small perturbations. On the other hand, other neural models may be able to more faithfully capture temporal dynamics or consider additional collective statistics that could be incorporated in future work. Thus, our formulation is a theoretically extensible and experimentally tractable framework for predicting the effects of perturbations.

An important caveat regarding our predictions is that the model is based on neural activity that may reflect other types of activity [40]. Calcium levels may indicate not only behavioral actuation and modulation but also proprioception, efference copies, or sensory information. They may depend strongly on environmental constraints such as bead immobilization here. Then, the interactions that we recover between neurons would confound physical and apparent interactions. As a result, a statistical connection from *in silico* individual neuron to collective synchrony may not indicate a causal pathway. Though this is a problem common to studies that rely solely on measurement data, previous work shows that maximum entropy models can in some cases identify real physical interactions from correlations [84]. A definitive answer, in any case, needs a perturbative experiment, and our formulation provides a set of predictions that can be directly checked.

To arrive at such verification, a perturbation experiment requires innovations on top of previous experiments. For example, small perturbations require a sufficient number of samples for the perturbed parameters to be measurable (Appendix I in S1 Text). By using the fact that the Fisher information is proportional to the number of independent samples and the Cramér-Rao bound, a lower bound on the number of experimental observations $T$ required to distinguish a change in the mode for a perturbation strength $\epsilon$ for Fisher information $F$ is on the order of

$$T \sim \left(\epsilon^2 F\right)^{-1}. \tag{8}$$

Taking a relatively large perturbation of $\epsilon = 10^{-2}$, we have $T \sim 10$ independent samples for an eigenvalue $\lambda \sim 10^3$, the order of magnitude of the largest mode (Appendix D in S1 Text). Linear growth in the number of required samples, however, restricts us from measuring insensitive modes in a reasonable amount of time (the experiments analyzed here last 8 minutes and collect roughly 80 to 120 independent samples [48]). Though mapping the full local

information geometry of $N = 50$ neurons would be difficult because the number of pairwise perturbations exceeds $\sim 10^6$ (accounting for two distinct pairs of matcher and target neurons), a similar experiment with about $N \sim 10$ neurons seems reasonable when coupled with techniques for incomplete matrix estimation [85]. Importantly, our formulation with pairwise perturbations and the maxent model strongly constrain both the high experimental and computational costs (further discussion in Appendix I in S1 Text).

Another limitation on implementation is that the duration of the experiment trials and the number of samples needed to estimate a maxent model limit the types of behaviors that could be studied. The experiments we consider last on the order of $10^2$ s, and so pairwise correlations can reflect neural activity on the timescale of several correlated behaviors including much faster head casts, bending, and even multiple locomotory reversals and the transitions between them [59, 86]. In this sense, perturbations indicate modifications to neural synchrony representative of this mixture and likely not of any single stereotyped behavior. In order to run our proposed protocol, it is important to specify time-averaged neural statistics of interest and to seek out recurrences of the same statistics at later points of observation. In other words, our modeling approach is flexible with respect to the particular statistics, but that choice must be made consistently throughout the experiment (Appendix I in S1 Text).

The feasibility of such an experiment stems from our argument that perturbative experiments harness analyticity to vastly simplify the range of possible perturbations. We exploit this property to extract a basis for the local information geometry including multi-component perturbations. One goal of such experimental intervention is then not to be more precise but sufficiently varied to span the local basis, an idea that can be generalized to other experimental systems besides the *C. elegans* model we consider.

Our procedure is general enough to be applied to other neural systems and collective behavior in systems biology. Experimentally, our method requires measuring and making small perturbations to neural activity at the level of individual components, along with the simultaneous measurement of a collective state or output behavior. Theoretically, it relies on a statistical modeling approach that assumes a discrete, time-independent description. One could imagine adapting the method to hormone concentration, gene expression, or other experimentally accessible quantities [87–92] and to output behavior including morphological descriptors [42, 93, 94], a stereotyped behavioral sequence [50, 95, 96], or developmental outcomes. Of these possibilities, the most straightforward adaptation would be to other model neural systems such as cortical cell cultures or muscle neurons involved in *Drosophila* flight [97]. Although neurons in these systems show discrete action potentials instead of graded activity, the standard mapping to binary states could lead to comparable results. Similar modeling approaches to the aforementioned systems, such as by discretizing levels of gene expression, could lay the basis for further extensions of the approach. Furthermore, we focused on a collective statistic measured at the same short timescales as the neural perturbations, but this could be expanded to measurements over longer timescales coinciding with behaviors of interest. In this sense, our work provides a generalizable theoretical framework that also incorporates specific perturbative predictions that could be tested experimentally across multiple biological systems.

The information geometry of scientific theories more generally suggests that they show sparse structure, where a few parameter dimensions strongly change the qualitative characteristics of phenomena and most dimensions are unimportant. This is a result of the logarithmically spaced eigenvalues of the FIM, or "sloppiness" [44, 78, 98, 99]. Whether by nature or by design, such quantitative reduction vastly simplifies the level of detail required for approximate theories, allowing for accurate prediction even with large uncertainty in most parameters [100, 101]. We find here a variation on this idea in the statistics of neural activity in *C. elegans*.

While sensitivity is concentrated in a few modes, neural activity is not conventionally sloppy because the largest eigenvalues decay slower like Zipf's law. What explains the slower decay or self-similarity? Might this be a feature of biological neural networks to reduce the dimensionality of control parameters or to nest varying levels of control? The state-of-the-art today with single-neuron measurement in *C. elegans* and optical genetics is approaching a point where experiments to test this hypothesis may become a reality.

## Supporting information

**S1 Text. Discovering sparse control strategies in neural activity.** Appendices A-J. (PDF)

## Acknowledgments

We would like to thank the UNM Center for Advanced Research Computing, supported in part by the National Science Foundation, for providing high performance computing resources used in this work. We thank Matthew Fricke for invaluable assistance with these resources. We thank the Santa Fe Institute for providing computational resources used for this work. The computational results presented have been achieved in part using the Vienna Scientific Cluster (VSC). We acknowledge K. Hallinen's help with their data repository. X.C. acknowledges Francesco Randi for insightful discussion.

## Author Contributions

**Conceptualization:** Edward D. Lee.

**Data curation:** Xiaowen Chen.

**Formal analysis:** Edward D. Lee, Xiaowen Chen, Bryan C. Daniels.

**Funding acquisition:** Edward D. Lee, Xiaowen Chen, Bryan C. Daniels.

**Investigation:** Edward D. Lee, Xiaowen Chen, Bryan C. Daniels.

**Methodology:** Edward D. Lee, Xiaowen Chen, Bryan C. Daniels.

**Project administration:** Edward D. Lee.

**Resources:** Edward D. Lee, Xiaowen Chen.

**Software:** Edward D. Lee, Xiaowen Chen, Bryan C. Daniels.

**Supervision:** Edward D. Lee.

**Validation:** Edward D. Lee, Xiaowen Chen, Bryan C. Daniels.

**Visualization:** Edward D. Lee, Xiaowen Chen, Bryan C. Daniels.

**Writing – original draft:** Edward D. Lee.

**Writing – review & editing:** Edward D. Lee, Xiaowen Chen, Bryan C. Daniels.

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
