## [Decision Letter · Decision Letter 0]

17 Nov 2021

Dear Dr. Lee,

Thank you very much for submitting your manuscript "Discovering sparse control strategies in C. elegans" for consideration at PLOS Computational Biology.

As with all papers reviewed by the journal, your manuscript was reviewed by members of the editorial board and by several independent reviewers. In light of the reviews (below this email), we would like to invite the resubmission of a significantly-revised version that takes into account the reviewers' comments.

We cannot make any decision about publication until we have seen the revised manuscript and your response to the reviewers' comments. Your revised manuscript is also likely to be sent to reviewers for further evaluation.

Sincerely,

Matthieu Louis

Associate Editor

PLOS Computational Biology

Lyle Graham

Deputy Editor

PLOS Computational Biology

Reviewer's Responses to Questions

**Comments to the Authors:**

Reviewer #1: My review is also attached in PDF format for readability.

Review PCOMPBIOL-D-21-01485

Discovering sparse control strategies in C. elegans

The authors leverage a maximum entropy (max-ent) model of the neural activity of the nematode C. elegans to probe the sensitivity of some collective neural statistics to experimentally inspired in-silico perturbations. Calcium recordings of neural dynamics are first discretized into 3-states, and a topologically-conserving maximum entropy model is built to match pairwise correlations and mean activity. Given this class of models, the authors choose a fine-grained statistical measure, termed “synchrony” (the number of neurons in each of the 3-states) and probe the sensitivity of this measure to small perturbations to pairs of neurons. The perturbation consists of clamping a matcher neuron to a target neuron with a small probability. Sensitivity of “synchrony” to such perturbations is measured in information-geometric terms through the Fisher Information Matrix (FIM) and its eigenvalues and eigenvectors. Through this methodology, the authors show that the “synchrony” is sensitive to a few principal perturbation modes, with the eigenvalue spectrum exhibiting a power law behavior for a significant number of modes revealing a wide hierarchy of “stiff” directions. In addition, the authors find that the large eigenvalue modes tend to be dominated by a small set of “pivotal” neurons, which tend to consistently increase or decrease the synaptic connections with matcher neurons. Given these findings, the authors argue that control of collective neural statistics is sparse in the number of neurons, while also being non-sloppy in the sense that the eigenvalues of the FIM decay initially as a power law and thus the system does not exhibit a clear separation between sloppy and stiff perturbative eigenmodes.

In summary, I am sympathetic with the general framework of the manuscript. Combining maximum entropy modelling with information-geometric analysis is somewhat innovative, and the authors show that it provides novel insight into the control mechanisms of biological neural networks, in a system that has already been extensively studied. However, besides some concerns regarding the legibility and the structure of the manuscript, I have some deep concerns regarding the validity and the generalizability of the approach that I think need to be addressed before I can recommend publication in this journal.

1) The analysis is based on a discretization of the neural dynamics that, while motivated by previous work, might miss important finer scale information. Indeed, unlike the spiking neurons of the mammalian brain, most C. elegans neurons exhibit continuous dynamics, and therefore the 3-state discretization might erase important information. The authors should better motivate this choice, also to help with the generalizability of the approach. Given time series data, how should one discretize before building a max-ent model?

a) I understand that building a maximum entropy model on a system with more states is challenging from a computational perspective, but I believe that this point should be explicitly discussed, especially considering the interpretation of the results.

2) It should also be clearly stated that the analysis is focused solely on time-invariant statistics of a 3-state discretization of the neural code. On this point I have a few detailed concerns:

(a) Maximum entropy models are typically used to infer the interaction rules that give rise to the observed steady-state distribution (given some underlying dynamics). Therefore, an important assumption underlying the inference of the model is that the system has been observed for a time scale longer than its relaxation time, such that it has clearly reached a steady-state. However, in the analyzed data it is unclear whether the observed statistics truly reflect the steady-state of the system, given that the measurement time, T, is relatively short compared to the transition time scales reported in Scholz et al.

(b) Neurons are inherently dynamical, and thus control mechanisms are also fundamentally dynamical, acting instantaneously on the continuous neural signal. In that sense, the max-ent model used in this manuscript can only probe control at the level of steady-state statistics, and not fine scale control mechanisms.

(c) This is particularly relevant when justifying the use of the maximum-entropy model in an experimental protocol. Real-time perturbations are instantaneous, and measuring their impact through “synchrony” is only possible after a time scale comparable to the time scale used for inferring the maximum entropy model. Relating back to concern a.), it’s not obvious that splitting the data in two T/2 segments, for example, would not result in significant changes in the inferred max-ent model, even though there are no apparent external perturbations to the system.

3) While it has been shown (Chen et al. PRE 2019 2019;99(5):052418) that max-ent models can accurately approximate higher order correlations between neural states in C. elegans, there is no figure in the current manuscript that shows that that is true in this dataset. The reference given in Chen et al. for the data is a biorxiv preprint (Scholz et al. 2018, https://www.biorxiv.org/content/10.1101/445643v1), which has received contestation by other experts in the field in the comments section of bioRxiv. Indeed, to my knowledge the peer-reviewed version of the referenced manuscript is (Hallinen et al. eLife 2021;10:e66135), and not the reference given by the authors. Thus, I’d recommend that the authors clarify both the source of the data and the capability of the max-ent model to predict higher order statistics in this dataset.

4) The reasoning behind using subgroups of N=50 neurons should be more clearly explained in the current manuscript. One must resort to reading Chen et al. to know that it is related to overfitting.

5) The use of collective 3-state neural statistics as a proxy for behavior is imprecise and unjustified. Behavior is inherently dynamical in nature, exhibiting a continuity of time scales. In C. elegans specifically, behavior ranges from muscle contractions to single body waves, to sequences of body waves that give rise to forward, backward and turning bouts, to sequences of bouts that result in different navigation strategies and so on. On what level is the synchrony measure presented by the authors related to behavior? Specifically, the calcium activity of certain C. elegans neurons has explicitly been associated with the initiation of different bouts, like forward or backward locomotion, but longer time scale sequences of behaviors have yet to be associated with neural dynamics explicitly. In addition, given the inability of the max-ent model to capture neural dynamics, longer time scale sequences of behavior are inherently unattainable in the current approach. If the relationship between the chosen statistic and behavior is not clearly justified, I’d recommend that terms like “neural-behavioral map”, as used in the abstract, be removed.

6) Related to both points 3) and 5), the authors provide no justification for the use of a dataset with only two recordings on immobilized worms, even though the data provided by Hallinen et al. includes also other experimental protocols, including recordings in freely moving worms. What is the reason for excluding this data? This is especially puzzling given the authors’ claim that the presented approach can be generalizable to other “measures of internal states” including behavioral output. Could these statements be made precise? After all, simultaneous measurements of neural and behavioral dynamics are available, and I’d be curious to see how to translate this approach to behavioral dynamics.

7) The use of the synchrony statistic should be better justified. In what sense is this a complete enough measure of collective neural states and, more importantly, behavior? This is especially important given that all the results of the manuscript depend on this statistic.

a) How much more computationally expensive is it to include state identities into synchrony statistic?

b) How justified is the connection to the concept of sloppiness? Sloppiness is studied with respect to the likelihood function of a set of parameters given the data (Transtrum, J Chem Phys 2015;143(1):010901), which is not the same as choosing a test statistic and examining how it varies with respect to perturbations. Why is the FIM not computed w.r.t the likelihood function?

8) The authors make no attempt to connect to other modelling paradigms of collective neural activity in C. elegans, even though there are several examples that even connect to the underlying neurobiological control. Here’s a few examples, although there are others:

i) Brennan et al. eLife 2019;8:e46814

ii) Morrison et al., Front Comput Neurosci. 2021 Jan 22;14:616639

iii) Fieseler et al., J. R. Soc. Interface.1720200459

9) The Fisher Information Matrix computed in the manuscript measures the sensitivity of the chosen test statistic, to perturbations that are reflected in the changing of model parameters. In that sense, it seems to be specific both to the model class and the chosen statistics. How generalizable is this? Can this FIM analysis be reproduced using other modelling paradigms?

Besides these more general concerns, I also have an extensive list of more technical comments:

1) There is a typo on line 16: neruons should read neurons.

2) In the paragraph starting in line 18, environmental conditions should be included as an important barrier to the connection between global neural state and behavior. Indeed, this study uses experiments in which worms are immobilized and drugged, which severely impairs the mapping between neurons and behavior.

3) I really cannot see the relevance of the analogy with sensory processing (line 30) in the context of the neural-behavioral map, behavior is very different from perception. Can you clarify this point?

4) The statement in line 43 starting with “In the formulation” is unclear and grammatically incomplete.

5) Reference 48 in line 104 is not very precise in the context of behavioral sequences, specialy for C. elegans. While discrete behavioral states are defined in ref. 48, no explicit temporal sequences of behavior are described. In the context of C. elegans other references would be more suitable, e.g. (Gomez-Marin et al., J. R. Soc. Interface 13: 20160466) or more recently (Costa et al., arXiv:2105.12811v2).

6) How exactly are the states “flat”, “increasing”, “decreasing” defined from the calcium signals? Information about the discretization of the data should be explicitly included in the methods. The reader shouldn’t have to resort to other references for such important information.

7) In line 146, reference to figures A6 and A7 is unjustified. Where is there a figure to support the fact that accounting for pairwise correlations is sufficient to capture higher-order correlations? Also, at this stage the statistic phi_coarse is not defined and perhaps should be defined in the figure caption.

8) There is no reference in the text of the manuscript to Fig.3. Also, it is not specified whether phi_fine or phi_coarse are used in Figs.3C,D,E,F,G. In addition, Fig 3D is very hard to read.

9) The section about the different types of perturbations starting in line 188 should be made clearer and perhaps there should be a reference to Appendix D.

10) The reference to Fig. C.10 in line 234 is challenging to understand as the exponents have not been defined yet.

11) In lines 236 and 237, it’s not at all clear where the Z_max values are coming from, and Fig. E.19 is not clarifying that. There should be more guidance to the figure or in the figure captions to make these points clearer.

12) In the paragraph before Eq.(8), how are the diagonal blocks exactly defined? Is it possible to show a figure of these matrices and how the blocks are identified?

13) In Eq.(8), how exactly are the eigenvectors normalized?

14) The shuffle models are not clearly explained in the text, and so their relevance becomes difficult to judge. What is the point of comparing with the different shuffles? There should be a section explaining these in detail and their relevance. In particular, the shuffled couplings control is confusing as it seems to only shuffle the neuron indices but preserving everything else. What’s the point?

15) How surprising is it that the pivotal neurons fluctuate across random subsets, given the fact that the statistic that is used to compute the FIM is independent of state and neuron identity?

a) Relatedly, it should be possible to reliably obtain neuron identities, even if only from a subset of neurons. Certain neurons are known to be important for initiating and maintaining certain behaviors (e.g. AVA), where are these in your analysis? Do the most consistent pivotal neurons identified in your approach correspond to known important neurons for C. elegans behavior? It would be very interesting if you could establish any testable predictions regarding the underlying neurobiology.

16) In Fig. 4A, the inset matrices are hard to see. I’m recommend expanding them.

17) The discussion paragraph starting in line 320 is also confusing. Doesn’t shuffling the interactions effectively give neurons different roles?

18) In the second paragraph of Appendix A there is a typo. Before references [90,91], “of” in “of the likelihood” should be removed.

19) It is difficult to see, comparing Fig. C14 with Fig. C11, that the distinction of pivotal neurons from uniformities with the MCH model is not as consistent as with the topological max-ent model introduced previously. Can this be explicitly quantified?

20) In Fig A.6C, the figure caption for the inset seems to be wrong.

21) The results of Fig.B.8 are insufficient to show that the eigenvectors of the inferred FIMs are robust to the sample size K. In fact, there seem to be significant differences in the secondary eigenvector between K=10^5 and K=10^6. Such differences should be better quantified, and this analysis should be done across multiple samples and not just one example.

22) There is a collection of typos in the first sentence of Appendix C.

23) Across the supplementary figures subsets A,B,C and D are used. How exactly are these subsets defined?

24) There is a typo in the last sentence of page 24: “and” in “consider and the fine” should be removed.

25) Some supplementary figures have been given the wrong labels. For instance, Fig. E.17 pertains to Appendix D but has been labelled as if belonging to Appendix E, and Figures G have nothing to do with Appendix G. The labelling is confusing.

26) “Furthermore” is repeated in two subsequent sentences in the first paragraph of Appendix G.

Reviewer #2: Summary of the paper:

The authors of the manuscript “Discovering sparse control strategies in C. elegans” describe an analytical perturbation protocol that is potentially experimentally feasible and should yield insight into the neurons that are most relevant to behavior. They use a (published) maximum entropy model extracted from the measured neuronal activity of C. elegans to test the impact of their pairwise perturbations in this system, finding that some specific ‘pivotal’ neurons are very sensitive to perturbation, possibly indicating that these are also relevant for behavior.

Review summary:

While I think the work itself is interesting and a relevant addition to the field, adding rigorous theoretical underpinnings to perturbation experiments that are now feasible or at least within reach, I have some major concerns about this paper which I detail below. Foremost, these are concerned with the presentation and context of this work, rather than the theoretical results. I therefore believe these are addressable by a major revision and editing for clarity.

Major issues:

There are some inconsistencies in the framing of the paper: the authors claim in the introduction and the abstract that this perturbation protocol is “(...) for systematic perturbation of neural states that limits experimental complexity but still characterizes collective aspects of the neural-behavioral map.”. The introduction refrains from considering any of the non-feedforward nature of neural systems where not all neural signals are directly mapping to behavior. Representations of behavior might be due to motor commands, or sensory-motor transformation. But they might also reflect proprioception or efference copies. The connection from neural activity to behavior and the concrete impact of this work needs to be better explained, especially since the maxent model is based on calcium imaging data.

A key relevant citation for high-throughput optogenetics in C. elegans is missing: https://doi.org/10.1038/s41592-018-0233-6 which also finds a small number of locomotion-relevant neurons in the worm

The introduction mixes organisms, techniques and results across a number of (very different) organisms to support their points. For example in ln.13-17, and especially ln. 58 where Phineas Gage and worms find themselves in the same citation. The motivation could be strengthened significantly by selecting citations from the same or similar organisms that are technically available for these measurements or alternatively expanding the discussion of the other results substantially and how these affect the interpretation with respect to the proposed perturbation approach.

There exist a number of similar, published datasets from other groups (e.g. from the Zimmer lab). Do the results generalize to datasets obtained with slightly different imaging modalities? If not, why?

Clarity of text and presentation: Panels and one figure seem to not be cited in the main text (e.g., Figure 3, 4B). It is also a bit confusing when large sections of the text refer to figures that are relegated to the appendix, this is mostly the case in the section about the Eigenvalue decay statistics. I think the work could benefit by a thorough editing for clarity in text and visual presentation. The appendix is clearly written and more approachable than the main text, and the main text would benefit from a similar tone (e.g Appendix A and C are particularly well done). I wonder if the authors would consider compressing the main text even more and focusing on the main argument of the perturbation strategy and the results for the C. elegans datasets, to make this manuscript more easily readable and a useful resource for different readers.

In the calculation often a subset of N=50 neurons is used, when the nervous system of the worm comprises 302 neurons. Does the fact that in this subset some coupling are missing affect the conclusions, especially the conclusion about sparse control? If not, then do we learn something about an efficient experiment where one only needs to observe a few neurons?

I am curious to know if the fact that the model is derived from calcium imaging data changes the interpretation of the model at all. It seems that calcium can detect neural activity easier than suppression such that there is a fundamental asymmetry in the underlying data. How does this impact the ability to detect certain couplings, and relatedly the column uniformity in Fig.4?

I am a bit confused by the following statement: ln 285-286: “In contrast with the typical notion that the identities of particular neurons are essential, pivotal neurons fluctuate between subsets and random samples as shown in Figure 5.” This seems to contradict much of the literature in which neurons do play distinct roles in behavior and these have been identified in the worm e.g. AVAs roles in reversals. It would be helpful for the reader to unpack this and similar statements about the sloppy-sparse control and put them in the context of the real, biological network. This would add to the discussion.

Minor:

Paragraph ln. 93-106 missing citation 10.1016/j.cell.2020.12.012

Citation [48] should instead cite one or more of the many C. elegans behavioral papers that are relevant here

Panel 4C: the maxent model data is barely visible

Check panel labelings 4

Appendix C ‘...turning them up, for a particular neuron is effective’...effective in what way?

Reviewer #3: The authors develop a theoretical framework using perturbation experiments to discover the neurons in a nervous system that induce the most change in the behavior of the system. They apply their methodology to a minimal model of C. elegans neural activity, and they find the model is most sensitive to a few principal perturbative modes.

One of the primary contributions of this work is the way in which extends earlier work on “sloppy” models. Whereas the original work primarily relies on an analysis of the curvature of the manifold from the dynamical description of the system, this work estimates the curvature through local perturbations. This is of practical use to neuroscientists who can only perturb their model living organisms.

My main issue with this paper is that it is hard to follow. This is partly because the approach is itself relatively sophisticated. But part of the difficulty is due to the writing and the organization.

According to the title, the expectation is that there is a certain focus on learning something specific about C. elegans. However, the C. elegans model as an example system diverts from the real focus of the contribution, which is the framework. The paper’s main contribution does not seem to be about the discovery of sparse control strategies in C. elegans. Instead, the paper’s central contribution seems to be about the framework for discovering the neural components that affect behavior, and therefore are likely to be involved in the control of the nervous system.

Presumably the choice of a C. elegans model was to provide useful predictions for experimentalists. However, the predictions are not clear, particularly in relation to understanding the neural basis of C. elegans behavior. And unfortunately, the choice of C. elegans model leaves the validity of the framework untested. Given that the focus was on showing the utility of their novel perturbative approach, a simple toy neural-behavioral model, where the ground-truth would be known and tractable, would have made the approach both more convincing, and more easily interpretable.

My suggestion would be to improve the clarity of the contribution by either making the focus more clearly on C. elegans and elaborating more succinctly on the predictions, or to focus more concretely on the framework and to provide first a toy model that illustrates the usefulness and validity of the approach, before applying it to data from C. elegans.

**Have the authors made all data and (if applicable) computational code underlying the findings in their manuscript fully available?**

Reviewer #1: **No: **The dataset had been previously published (although the reference provided in the current manuscript is wrong, see Comments to the Authors), but no explicit link to the data used in this manuscript is provided. Also, the code available on github, but currently there is no section in the main text or appendix pointing to the code. I believe that this information should be stated more explicitly, with direct links to both code and data appearing on the main text (or in Appendix).

Reviewer #2: Yes

Reviewer #3: Yes

PLOS authors have the option to publish the peer review history of their article (what does this mean?). If published, this will include your full peer review and any attached files.

Reviewer #1: No

Reviewer #2: No

Reviewer #3: No
---

## [Decision Letter · Decision Letter 1]

18 Mar 2022

Dear Dr. Lee,

Thank you very much for submitting your manuscript "Discovering sparse control strategies in neural activity" for consideration at PLOS Computational Biology. As with all papers reviewed by the journal, your manuscript was reviewed by members of the editorial board and by several independent reviewers. The reviewers appreciated the attention to an important topic. Based on the reviews, we are very likely to accept this manuscript for publication, providing that you modify the manuscript as best as possible according to the remaining recommendations of reviewers #1 and #2.

Sincerely,

Matthieu Louis

Associate Editor

PLOS Computational Biology

Lyle Graham

Deputy Editor

PLOS Computational Biology

[LINK]

Reviewer's Responses to Questions

**Comments to the Authors:**

Reviewer #1: The review is uploaded as an attachment.

Reviewer #2: The authors have addressed my comments.

I have a few remaining textual changes to improves the accuracy of the terms used.

"making small perturbations to internal states at the level of individual

components"

Internal states has a distinct meaning in neuroscience, e.g, motivation, hunger, sleepiness ... which could be confusing to the reader in this sentence. The authors should just use neural activity in its place here.

Later in the same paragraph, the authors even refer to that meaning of 'internal state', but also use gene expression in the list. I would recommend staying with 'microscopic quantities' or similar to avoid any confusion.

"the anterior neural network" - this should be anatomically correctly named as there is no such thing as an anterior neural network in C. elegans: a subset neurons from the anterior ganglia in the worm head or

simpler 'neurons located in the anterior of the worm'

Minor: References to “Theoretical formulation.” should be replaced with the new section title

**Have the authors made all data and (if applicable) computational code underlying the findings in their manuscript fully available?**

Reviewer #1: Yes

Reviewer #2: Yes

PLOS authors have the option to publish the peer review history of their article (what does this mean?). If published, this will include your full peer review and any attached files.

Reviewer #1: No

Reviewer #2: No

Figure Files:

Data Requirements:

Reproducibility:

References:

---

## [Editor Report · Decision Letter 2]

1 Apr 2022

Dear Dr. Lee,

We are pleased to inform you that your manuscript 'Discovering sparse control strategies in neural activity' has been provisionally accepted for publication in PLOS Computational Biology.

Best regards,

Matthieu Louis

Associate Editor

PLOS Computational Biology

Lyle Graham

Deputy Editor

PLOS Computational Biology

---

## [Editor Report · Acceptance letter]

3 May 2022

PCOMPBIOL-D-21-01485R2 

Discovering sparse control strategies in neural activity

Dear Dr Lee,

I am pleased to inform you that your manuscript has been formally accepted for publication in PLOS Computational Biology. Your manuscript is now with our production department and you will be notified of the publication date in due course.

With kind regards,

Zsanett Szabo
